# DAG-MoE: From Simple Mixture to Structural Aggregation in Mixture-of-Experts

## Abstract

Mixture-of-Experts (MoE) models have become a leading approach for decoupling parameter count from computational cost in large language models. Despite significant progress, effectively scaling MoE performance remains a challenge. Previous work shows that the use of fine-grained experts enlarges the space of expert combinations and can improve flexibility, but it also imposes substantial routing overhead, creating a new scalability bottleneck. In this paper, we explore a complementary axis for scaling — expert-output mixture. We first analyze the limitations of the standard weighted-summation aggregation in conventional MoE architectures. We then theoretically demonstrate that introducing structural aggregation both expands the expert-combination space without altering the experts or router configuration and enables possible multi-step reasoning within a single MoE layer. To this end, we propose DAG-MoE, a sparse MoE framework that employs a lightweight module to automatically learn the optimal aggregation structure among the selected experts. We evaluate DAG-MoE under standard language modeling settings. Extensive experiments show that DAG-MoE consistently improves performance in both pretraining and fine-tuning, surpassing traditional MoE baselines.

## 1 Introduction

Mixture-of-Experts (MoE) models (Shazeer et al., 2017b; Lepikhin et al., 2020; Fedus et al., 2022) have recently emerged as a state-of-the-art architecture for large-scale foundation models, such as Large Language Models (LLMs). Compared to standard dense models, MoE decouples model size from computational cost by splitting a large dense neural network into multiple smaller experts, with a router dynamically selecting the top-$K$ most relevant experts for each input. This paradigm has been widely adopted in recent open-source LLMs and multimodal models (Liu et al., 2024; Yang et al., 2025; Meta, 2025; Muennighoff et al., 2025; Li et al., 2025b).

Despite the success of MoE, how to effectively scale their performance remains unclear. Prior works have examined various aspects: some studies improve the routing mechanism by proposing advanced routing algorithms (Zhou et al., 2022; Qiu et al., 2024; Chi et al., 2022; Wang et al., 2024); others investigate the relationship between MoE performance and sparsity (Li et al., 2025a; Tian et al., 2025b); and another line of research explores the effect of expert granularity (He, 2024; Ludziejewski et al., 2024). In particular, these studies show that MoE performance can be improved by increasing both the total number of experts and the number of active experts while reducing the size of each individual expert. While this strategy has strong potential, it also introduces substantial router-side complexity, creating a new bottleneck for scaling. Consequently, state-of-the-art MoE systems typically avoid extremely fine-grained configurations. Nevertheless, most existing work on MoE overlooks a critical component of the architecture: how expert outputs are mixed. In standard MoE, once the router selects the top-$K$ experts, the final representation is formed by a weighted sum using the router scores as coefficients. Because weighted summation is permutation-invariant, the output is fully determined by the set of selected experts—independent of their ordering or interactions—thereby constraining the framework's flexibility and expressiveness.

In this paper, we propose **replacing simple mixing with structural aggregation** in MoE. Specifically, we organize the aggregation of the top-$K$ experts into a directed acyclic graph (DAG), assigning each expert a distinct structural role within the graph. output of experts are then aggregated

according to the prescribed graph structure. We demonstrate that introducing structural relationships among experts effectively expands the expert combination space without modifying the experts or the router, and avoids adding additional complexity on the router side, thereby increasing the model's expressiveness. Furthermore, the DAG structure enables the approximation of multi-step reasoning within a single MoE layer, which is beneficial for solving complex problems such as dynamic programming. To this end, we present DAG-MoE, which incorporates a DAG learning module within the MoE layer that automatically discovers the optimal structure among the selected top-$K$ experts and performs aggregation according to that structure through iterative refinement. We conduct extensive experiments on both language modeling tasks and downstream tasks, and DAG-MoE consistently outperforms standard MoE architectures, demonstrating the effectiveness of structural aggregation relative to simple mixing.

## 2 PRELIMINARIES

**Architecture of MoE.** In the paper, we consider standard transformer-based LLM model. Let $x \in \mathcal{R}^d$ denote the input token embedding in LLM. Let $\{E_i(\cdot)|i = 1 \ldots, N\}$ be a set of $N$ expert networks, where each expert is an identical Feed-Forward Network (FFN) with inner hidden size $d_r$. Finally, let $g_i(\cdot)$ be the sparse gate function in router that choose the top-$K$ experts given the input token. The output of the MoE layer is computed as:

$$y = \sum_{i=1}^{N} g_i(x)E_i(x),\tag{1}$$

where the gating function is defined as:

$$g_i(x) = \begin{cases} s_i, & s_i \in \text{TopK}(\{s_j\}_{j=1}^{N}, K), \\ 0, & \text{otherwise}, \end{cases} \quad \text{and} \quad s_i = \delta(e_i^\top x).\tag{2}$$

The $\delta$ is the score function, which can be implemented by the Softmax or the Sigmoid function. $e_i \in \mathcal{R}^d$ is a learnable vector related to $i$-th expert. The gating function ensures that only the top-$K$ experts can be used to compute the output of token $x$ in the MoE layer.

**Granularity of MoE.** Key hyperparameters of an MoE model include the number of experts $N$ and the number of active experts $K$. Recent studies show that increasing the granularity of experts can scale up the performance of MoE (He, 2024; Ludziejewski et al., 2024). Specifically, granularity is defined as $G = \frac{d_f}{d_r}$, where $d_f$ is the hidden size of a dense FFN counterpart. For example, suppose $d_f = 2048$ and $d_r = 1024$, the granularity is 2 and we pick top-2 experts. Holding $d_f$ fixed while decreasing $d_r$ increases $K$ (and also $N$, as we typically assume the total parameters in the MoE layer are fixed). Higher granularity enlarges the space of possible expert combinations and thus the model's flexibility and expressiveness. For example, suppose that we have 8 experts and we choose the top-2 experts, there are in total $\binom{8}{2} = 28$ different combinations. Instead, if we have 16 experts and we choose the top-4 experts, the total combination quickly increases to $\binom{16}{4} = 1820$. Despite these benefits, fine-grained MoE introduces additional routing complexity. First, as each expert $i$ needs a learnable vector $e_i$ to serve as a router key, increasing the number of experts greatly expands router parameters. Moreover, identifying the optimal expert set per token becomes harder, and maintaining balanced token–expert assignments further complicates training. Substantial efforts have aimed to improve fine-grained MoE. PEER (He, 2024) reduces expert vector size via product-key retrieval Lample et al. (2019), but it remains costly when scaled up to millions of experts. (Ludziejewski et al., 2024) uses an expert-choice router to address balance issues, but this proves suboptimal compared to token-choice routers in large-scale training (Muennighoff et al., 2025).

## 3 DAG-MoE: HARNESSING THE POWER OF STRUCTURE IN MoE

### 3.1 FROM SIMPLE MIXTURE TO STRUCTURAL-GUIDED AGGREGATION

Given the current limitations of MoE scaling, we explore alternative ways to enhance MoE capability. We begin by discussing the expressiveness of the standard MoE architecture. As shown in Eq. 1,

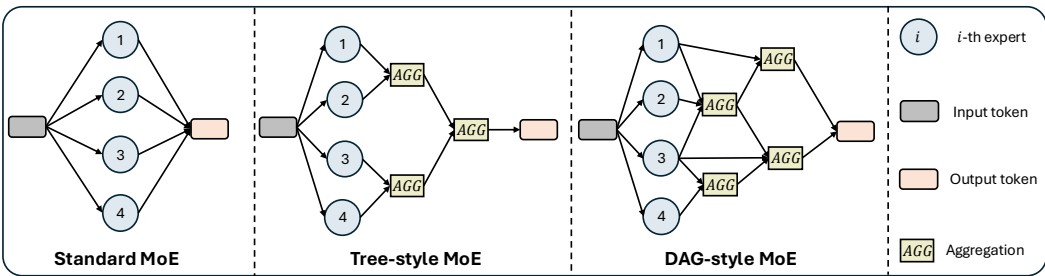

Figure 1: Comparison of different mixing structures in MoE.

expressiveness is determined by two components: the expert set $\mathcal{F} = \{E_i(x) \mid i = 1, \ldots, N\}$ and the score set $\mathcal{S} = \{g_i(x) \mid i = 1, \ldots, N\}$. This can be broken down into the capacity of each $E_i(x)$ and $g_i(x)$, and the cardinalities $|\mathcal{F}|$ and $|\mathcal{S}|$. Each $g_i(x)$ is a single scalar, thus having constant expressiveness, while each $E_i(x)$ is an FFN with identical architecture whose capacity is determined by its parameter size. Since increasing expert size raises practical concerns, throughout the discussion, we assume that the total parameter budget for $\mathcal{F}$ remains fixed. Under this constraint, improving expressiveness depends on scaling the cardinality of $\mathcal{F}$ and $\mathcal{S}$, which both equal to $N$. In essence this equivalent to increase the granularity of MoE. Once both $\mathcal{F}$ and $\mathcal{S}$ are determined, the overall expressiveness of the MoE is bounded by the functional form of Eq. 1, which is also fixed once the top-$K$ experts are chosen by the router, due to the permutation-invariant nature of summation. To further improve the model capacity, a natural question arises:

**_Is there a more effective way other than summation to combine information from the experts?_**

Let us view the experts from a structural perspective and consider the case where the top-4 experts are selected. As shown on the left of Fig. 1, the aggregation process in standard MoE can be interpreted as a computational graph in which each expert corresponds to an isolated node with no edges. In this setting, permuting the order of experts does not change the graph, since all nodes share the same structural role. Now, let $AGG$ denote the aggregation function, and consider a tree-structured computation over the top-4 experts. As illustrated in the middle of Fig. 1, the experts are arranged into a hierarchy of depth 2: at the first level, experts 1 and 2 are combined by $AGG$ to produce an intermediate representation, while experts 3 and 4 are simultaneously merged by another instance of $AGG$. At the second level, these two intermediate outputs are further combined using additional instances of $AGG$. In this setup, swapping experts 1 and 3 changes the final output, because the second-level operations now act on different inputs. This indicates that experts 1 and 3 occupy distinct structural roles within this expert graph. More generally, the top-4 experts can be organized into a directed acyclic graph (DAG), with an example shown in the right panel of Fig. 1. By simply permuting the four experts or choosing a different DAG, the computational graph changes and yields a different output. Thus, even without modifying the experts or the router, the expressiveness of the MoE architecture can be substantially enhanced through structural composition alone. For a given $K$, the number of possible DAGs grows exponentially, offering a vast space of structural configurations. We refer to this form of structural aggregation as **DAG-style MoE**.

### 3.2 A GENERAL FORMULATION OF DAG-STYLE MoE AND THEORETICAL ANALYSIS

In this section, we formally define and analyze DAG-style MoE. We assume the expert and router configurations remain the same as in standard MoE, and denote the list of top-$K$ experts selected by the router as $\boldsymbol{k} = [k \mid s_k \in \text{TopK}(\{s_j\}_{j=1}^N, K)]$. We then define $\mathcal{G}(K)$ as the set of all possible DAGs constructed from $K$ experts, where each DAG is represented as $G = (\mathcal{V}, \mathcal{A})$. Here, $\mathcal{V}$ denotes the set of nodes, and each node $v \in \mathcal{V}$ corresponds to an output representation, which can be either an initial expert output or an intermediate result produced by the aggregation function $AGG$. Let each node be indexed by $v = (l, i)$, where $l$ denotes the depth of the node and $i$ its index at depth $l$. For example, the initial output from expert 1 corresponds to node $(0, 1)$. The set $\mathcal{A}$ is the adjacency list, which specifies the connections within the DAG. Formally, we define $\mathcal{A} = \{A_i^l \mid l = 1, \ldots, L; \ i = 1, \ldots, n(l)\}$, where $L$ is the maximum depth of the DAG and $n(l)$ denotes the number of nodes at depth $l$. Each $A_i^l$ is the set of incoming nodes connected to node $(l, i)$. Formally, $(k, j) \in A_i^l$ means that node $(k, j)$ connect to node $(l, i)$ and we have $k < l$. For example, the adjacency list of the tree graph shown in the middle of Fig. 1 can be written as $\mathcal{A} = \{A_1^1 = \{(0, 1), (0, 2)\}, A_2^1 = \{(0, 3), (0, 4)\}, A_1^2 = \{(1, 1), (1, 2)\}\}$.

Now, let $x_i^l$ denote the learned representation of node $v = (l, i)$. For a given DAG $G \in \mathcal{G}(K)$, the corresponding computation in DAG-style MoE can be formulated as:

$$x_i^0 = g_{\mathbf{k}[i]}(x) E_{\mathbf{k}[i]}(x), \quad i = 1, \ldots, K, \tag{3}$$

$$x_i^l = AGG(\{x_j^k | (k, j) \in A_i^l\}), \quad i = 1, \ldots, n(l), l = 1, \ldots, L - 1, \tag{4}$$

$$y = AGG(\{x_j^k, | (k, j) \in A_1^L\}). \tag{5}$$

Here, we assume that the last depth $L$ only have one node $(L, 1)$ and its connection in $G$ is defined by $A_1^L$. With the above formulation in place, we are now prepared to present a formal analysis and theoretical results for DAG-style MoE.

**Theoretical expressiveness.** Throughout the discussion, we assume $AGG$ is a sufficiently powerful injective function over set inputs, which can be readily implemented using MLPs combined with summation, min, or sum (Zaheer et al., 2017; Xu et al., 2018). Under this assumption, we show that DAG-style MoE possesses strictly greater expressive power than standard MoE.

**Proposition 3.1.** *Given a top-$K$ experts list $\mathbf{k}$, any DAG-style MoE satisfy Eq. 3-Eq. 5 can injectively encode any $G \in \mathcal{G}(K)$ if $AGG$ is injective.*

**Theorem 3.2.** *Given a top-$K$ experts set $\mathbf{k}$, any DAG-style MoE satisfy Eq. 3-Eq. 5 is strictly more powerful than standard MoE with Eq. 1 if $AGG$ is injective.*

We leave the detailed proof in Appendix A.1. Briefly speaking, we connect the DAG-style MoE to the massage passing graph neural network (Xu et al., 2018; Gilmer et al., 2017) and leverage the results from D-VAE (Zhang et al., 2019) to prove that the above formulation can injectively encode any different DAG structure and node permutation, while the standard MoE cannot, which is sufficient to prove that DAG-style MoE holds better expressiveness than standard MoE.

**Benefits for Reasoning.** Next, we discuss the practical benefits brought by the greater expressiveness of DAG-style MoE. Specifically, we use dynamic programming (DP) as a running example. DP is a foundational paradigm for solving decision-making and combinatorial optimization problems. We defer the formal definition of the general DP framework to Appendix A.2.1. In general, solving a DP problem entails iteratively solving a partially ordered collection of subproblems and aggregating their solutions to produce the final answer. Recent work shows that Transformers without a Chain-of-Thought (CoT) mechanism (Wei et al., 2022) cannot effectively solve dynamic programming (DP) tasks (Feng et al., 2023), as a constant-depth model cannot simulate the requisite multi-step subproblem computations. Importantly, many DP solution procedures induce a directed acyclic graph (DAG) over subproblems via their natural partial order, which suggests a direct fit for DAG-style MoE. Conceptually, due to the flexibility of the DAG-style MoE, a well-trained model can naturally learn a alignment between the DP problem and corresponding DAG solving process. By learning mapping between each input and answer of each subproblem through $AGG$ and perform aggregation according to the DAG structure, it can directly output the final solution of a DP problem. In what follows, we demonstrate that DAG-style MoE is much more effective than standard MoE to emulate this multi-step reasoning. Let $G(dp)$ be the computational DAG for a DP problem and $L(dp)$ be the maximum depth of the $G(dp)$, we have:

**Theorem 3.3.** *For any integer $n \in \mathbb{N}$, consider any DP problem satisfying Assumption 4.2 to 4.5 in (Feng et al., 2023) with problem size length less than $O(K \log(n))$, there exists a log-precision constant depth transformer composed of (i) multi-head attention block and (ii) at least one DAG-MoE block with top-$K$ experts and number of iteration $L \geq L(dp)$ and computation defined by $G(dp)$, with hidden dimension $d$ and parameter size bounded by $O(poly(n))$ that can output the correct DP answer.*

We defer the detailed proof to Appendix A.3. At a high level, by constructing the DAG-style MoE computation to mirror the DP computation graph $G(dp)$ the model can explicitly simulate all intermediate subproblem steps for DP instances of size $O(K \log(n))$ (with $K$ active experts), whereas standard MoE can realize only a single aggregation step, due to its permutation-invariant summation. While solving a full $O(n)$-sized DP instance in a single forward pass with a constant-depth DAG-style MoE remains challenging, a single layer can nevertheless execute multiple reasoning steps, effectively increasing the logical depth of a standard Transformer layer without significant parameter and compute overhead. Due to space limit, we defer detailed example and additional discussion to Appendix A.2.2.

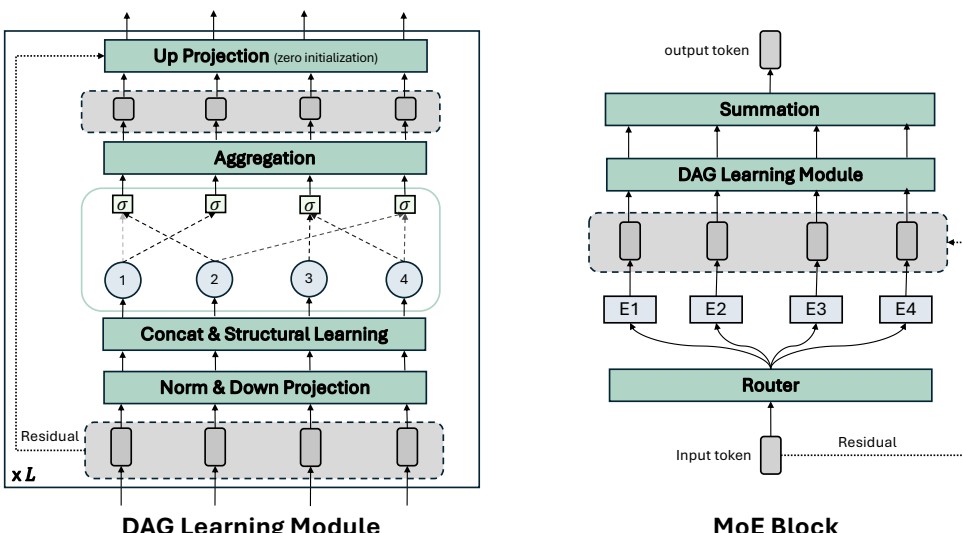

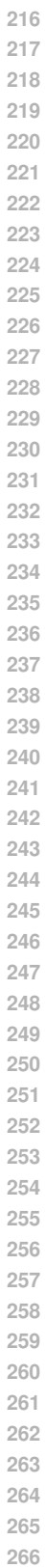

Figure 2: Left: the DAG Learning Module automatically learns an optimal DAG structure over the selected experts and execute DAG-style computation. Right: the complete MoE block in DAG-MoE.

### 3.3 DAG-MoE: LEARNING OPTIMAL DAG BETWEEN EXPERTS

As discussed in the previous section, organizing experts into a DAG offers several theoretical advantages. However, implementing a DAG-style MoE architecture is highly non-trivial. First, since the size of $\mathcal{G}(K)$ grows exponentially with $K$, the design space of possible DAG structures quickly becomes intractable. A straightforward solution is to predefine the structure and then design specific model tailored to the structure. For example, S′MoRE (Zeng et al., 2025) fixes the structure to a tree and employs a hierarchical router that selects experts at each depth in a top-down manner. However, this mechanism does not generalize to other structures, and different tokens may require different structures for optimal processing. In this section, we introduce DAG-MoE, a general and practical architecture that learns the optimal DAG structure and processes the aggregation accordingly.

**Architecture design.** Specifically, in DAG-MoE, given an input token, the router first selects the top-$K$ experts and obtains their output representations, which serve as the initial node set for DAG learning. A DAG learning module is then applied to infer the structure among these expert representations and perform aggregation based on the learned connections. This module operates for $L$ iterations, with each iteration applies the same module to learn structure at current depth, producing a DAG with maximum depth $L$. At each iteration $i$, node representations from the previous iteration are updated, and the connectivity for the current depth is determined. Formally, let $x_i^l$ be the output representation of node $i$ at depth $l = 0, \ldots, L$ for token $x$. The DAG learning module learns the DAG structure between experts and generates the final token representation through the following procedure:

$$x_i^{l=0} = g_{\mathbf{k}[i]}(x) E_{\mathbf{k}[i]}(x) + \frac{x}{K}, \quad i = 1, \ldots, K, \tag{6}$$

$$x_{i,\text{input}}^l = \text{LayerNorm}\left(x_i^{l-1}\right), \tag{7}$$

$$x_{i,\text{down}}^l = W_{\text{down}}^l x_{i,\text{input}}^l, \tag{8}$$

$$x_{(i,j)}^l = \text{Concat}(x_{i,\text{down}}^l, x_{j,\text{down}}^l), \tag{9}$$

$$e_{(i,j)}^l = \sigma\left(W_{\text{edge}}^l x_{(i,j)}^l\right), \quad \hat{x}_{(i,j)}^l = e_{(i,j)}^l * W_{\text{node}}^l x_{(i,j)}^l, \tag{10}$$

$$x_i^l = W_{\text{up}}^l \left( \sum_{j=1,\ldots,K} \hat{x}_{(i,j)}^l \right) + x_i^{l-1}, \tag{11}$$

where $W_{\text{down}} \in \mathbb{R}^{d_g \times d}$, $W_{\text{edge}} \in \mathbb{R}^{d_g \times 2d_g}$, $W_{\text{node}} \in \mathbb{R}^{d_g \times 2d_g}$, and $W_{\text{up}} \in \mathbb{R}^{d \times d_g}$ are learnable parameters and $d_g$ is the hidden size in DAG learning module, which can be independent to both the model hidden size $d$ and expert hidden size $d_r$. $\sigma$ is a nonlinear activation, which is implemented by the same activation function as in expert FFN. The final output of the MoE block is computed by:

$$y = \sum_{i \in 1,\ldots,K} x_i^L. \tag{12}$$

We now detail the process and key design choices in the DAG learning module. First, Eq. 6 computes the output representation for each selected expert $i$, serving as the initial node representation at depth 0. In addition, we incorporate the original token representation through a residual connection. To ensure that the total residual contribution remains normalized to 1 after summing over all experts in Eq. 12, the residual added to each node is scaled by a factor of $1/K$. Empirically, we find that both the residual connection and the scaling factor are crucial for stable optimization. To keep the DAG learning module lightweight and avoid excessive parameters, at each iteration the node representations are first normalized and then projected into a lower-dimensional space using Eq. 7 and Eq. 8. The structural relationships are then learned within this reduced space.

At iteration $l$, the structure is learned and executed as follows. First, we choose the number of nodes $n(l)$. In principle, it can be any positive integer, leading to an enormous search space. In DAG-MoE, we fix $n(l) = K$, which is equal to the number of experts. Next, for each node $(l, i), i = 1, \ldots, K$, we determine how it aggregates information from previous depths. Learning connections to all earlier nodes is computationally expensive and yields overly dense graphs. To mitigate this, we restrict each node $(l, i)$ to aggregate only from nodes at depth $l - 1$; information from depths $0, \ldots, l - 2$ is injected via residual connections. Finally, we use $x_{i,\text{down}}^l$ as the query for node $(l, i)$ and all $x_{j,\text{down}}^l, i = 1, \ldots, K$ for nodes $(l - 1, j)$. The connection between node $(l, i)$ and $(l - 1, j)$ are learned through Eq. 9 and Eq. 10.

Specifically, we first construct possible connections between $(l, i)$ and $(l - 1, j)$ for $j = 1, \ldots, K$ by concatenate $x_{i,\text{down}}^l$ with all $x_{j,\text{down}}^l$. Next, the edge is learned by the projection $W_{\text{edge}}$ and an activation $\sigma$. The edge can potentially carry various information, like the logic operation in reasoning tasks, or the relation in knowledge graphs, or even no connection. Next, the connection representation between node $(l, i)$ and $(l - 1, j)$ is learned by Eq. 10, which is essential an element-wise gating using $e_{i,j}^l$. Finally, the information is aggregated and projected back to the original dimension through Eq. 11, yielding the output representation for node $(l, i)$. Note that for the up projection, we use zero-weight initialization to stabilize training. The complete workflow of the DAG learning module, together with the MoE block in DAG-MoE, is depicted in Fig. 2.

**Computational cost analysis.** Since DAG-MoE introduces only the DAG learning module as an additional component, we focus our cost analysis on this module. Let the batch size be $B$ and the sequence length be $S$. The FLOPs of a single matrix multiplication are $2BSd_{\text{in}}d_{\text{out}}$. Based on this, the FLOPs of the DAG learning module for a single iteration are $\text{FLOPs}_{\text{dag}} = 4BSdd_g + 4 * K^2BS * 2d_gd_g = 4BSdd_g + 8K^2BSd_g^2$. In comparison, the FLOPs for an additional shared expert in MoE with hidden dimension $d_g$ is $\text{FLOPs}_{\text{expert}} = 4BSdd_g + 2dBSd_g$. Subtracting and dividing the common terms, the comparison reduces to $4K^2d_g$ versus $d$. In practical, as we have both $K \ll d$ and $d_g \ll d$, $4K^2d_g$ can be similar or smaller than $d$. However, there will be additional overhead for DAG learning module with multiple iteration due to the sequential nature of it.

## 4 RELATED WORKS

**Mixture of Experts (MoEs).** MoEs (Shazeer et al., 2017a; Fedus et al., 2022) have become a dominant paradigm for building large-scale models such as LLMs due to their remarkable computational efficiency. By decoupling model size from computational cost, the MoE design enables fine-grained control over inference efficiency. Many state-of-the-art LLMs adopt MoE architectures, including DeepSeek-V3 Liu et al. (2024), Mixtral Jiang et al. (2024), and Qwen-3 Yang et al. (2025).

**Architecture improvement of MoEs.** A number of works have sought to improve MoE architectures by modifying their core design. For instance, MH-MoE extends the MoE framework to a multi-head setting, analogous to the design of multi-head attention. CoE (Wang et al., 2025) introduces an iterative routing strategy, where the top-$K$ experts are selected through multiple rounds and the output representation is refined step by step. Recently, S'MoRE (Zeng et al., 2025) first introduced the notion of structural flexibility into the MoE architecture. Rather than selecting top-$K$ experts in a single step, S'MoRE employs a hierarchical router that iteratively selects experts across multiple stages. At each iteration, selected experts are connected to the next-layer experts through a non-linear transformation, forming a tree-like structure. The aggregation of expert outputs is then conducted based on this tree. While S'MoRE presents a novel structural perspective, it is primarily designed as a parameter-efficient adapter module for LLM fine-tuning and is not evaluated as a standalone MoE backbone. Moreover, its design remains limited to a tree-based structure and does not explore generalization to other structural forms.

**Scaling of MoEs.** Recently, a growing body of work has focused on understanding and improving the performance scaling of MoE architectures. For instance, OlMoE (Muennighoff et al., 2025) and Skywork-MoE (Wei et al., 2024) conduct extensive experiments to analyze different components and training strategies of MoE. Some studies improve the routing mechanism by proposing advanced routing algorithms, including expert-choice routing (Zhou et al., 2022), RNN routing (Qiu et al., 2024). Some also analyze the representation distribution between the router embedding and token embedding (Chi et al., 2022). Others try to improve the training stability by improving the balance loss of token-choice routing (Wang et al., 2024) or loss computation (Qiu et al., 2025). (Ludziejewski et al., 2024) study the effect of granularity and show that fine-grained experts improve the flexibility of MoE, thereby enhancing scaling performance. PEER He (2024) pushes the scalability of MoE to a million experts using product-key retrieval. (Abnar et al., 2025) investigates the optimal sparsity level for MoE architectures, while (Ludziejewski et al., 2025) examines the combined effects of factors such as data, model design, and training strategies. Finally, Ling-mini-beta (Tian et al., 2025a) evaluates the efficiency of MoE compared to dense models.

## 5 EXPERIMENTS

In this section, we empirically evaluate DAG-MoE. Specifically, we aim to answer the following questions: **Q1**: Does DAG-MoE exhibit greater performance than standard MoE across different base MoE configurations? **Q2**: How do the hyperparameters of the DAG learning module affect DAG-MoE 's performance? **Q3**: How does DAG-MoE perform on downstream language tasks? To address these questions, we integrate DAG-MoE as the primary MoE block in an LLM and train the model from scratch on casual language modeling task.

### 5.1 EXPERIMENTAL SETTINGS

We briefly discuss the setting of pretraining and fine-tuning, defer more details to Appendix B.

**Model Details.** The architecture of DAG-MoE is implemented on top of Llama3 (Dubey et al., 2024). Except the MoE block, we directly adopt the architecture and tokenizer of Llama3.1-8B. Due to resource constraints, we reduce the number of layers and the hidden size. For the MoE block, we use a standard token-choice router. The router follows the implementation of Switch Transformer Fedus et al. (2022) with standard balance loss. Meanwhile, we add Z-loss to regularize the learned router logits Muennighoff et al. (2025); Zoph et al. (2022). To systematically evaluate the behavior of DAG-MoE under different model size, we design three model variants: small, medium, and large, denoted as DAG-MoE-s, DAG-MoE-m, DAG-MoE-l, respectively. In the **DAG-MoE-s**, we use 4 layers with a hidden size of 512. For the MoE block, we further consider two settings: (1) a **coarse-grained** configuration with 32 experts per block, each expert being an FFN with $d_r = 256$, where the router selects the top-4 experts for each token; and (2) a **fine-grained** configuration with 64 experts per block, each expert having $d_r = 128$, where the router selects the top-8 experts per token. In the **DAG-MoE-m**, we increase the number of layers to 6. Finally, for **DAG-MoE-l**, we further increase the number of layer and hidden size to 8 and 1024. We also increase the expert size to $d_r = 512$. We keep the MoE block in the coarse-grained configuration for medium and large version. On top of the standard MoE block, we add the proposed DAG learning module and vary the $d_g$ and depth $L$. For the baseline, we adopt the same overall architecture as described above,

but replace the MoE block with a standard MoE. To ensure a fair comparison with DAG-MoE, we introduce a shared expert such that the total parameter count in the MoE block closely matches that of DAG-MoE.

**Data, training, and evaluation details.** We use the Pile dataset (Gao et al., 2020) as the pretraining corpus. We design two experimental setups. In the first experiment, we train DAG-MoE-s and DAG-MoE-m on approximately 12B tokens randomly sampled from the Pile and evaluate trained model on a held-out 1.3B-token subset. In the second experiment, we train DAG-MoE-l on about 40B tokens and evaluate on both in distribution and out-of-domain corpus, including Pile, FineWeb-edu (Lozhkov et al., 2024), Wikipeida text (Thrush et al., 2022), and C4 (Raffel et al., 2020). All models are trained with the standard causal language modeling objective (Radford et al., 2018) using a maximum sequence length of 2048. Perplexity on the evaluation sets serves as the primary metric.

**Fine-tuning setting.** To evaluate the downstream performance, we fine-tune DAG-MoE-l and its corresponding baseline, both pre-trained on 40B tokens. The fine-tuning datasets includes Alpaca (Taori et al., 2023), Open-Platypus (Lee et al., 2023), SlimOrca (Mukherjee et al., 2023), MathInstruct (Yue et al., 2023), Open-r1-math, and MetaMathQA (Yu et al., 2023). Models are trained for 3 epochs with a constant learning rate. After fine-tuning, we evaluate on downstream tasks including PIQA (Bisk et al., 2020), ARC-e (Clark et al., 2018), HellaSwag (Zellers et al., 2019), GPQA (Rein et al., 2024), Lambada (Paperno et al., 2016), MMLU (Hendrycks et al., 2009), and BBH (Suzgun et al., 2022). Detailed implementation and dataset description for fine-tuning are described in Appendix B.2.

## 5.2 PRETRAINING EVALUATION RESULTS

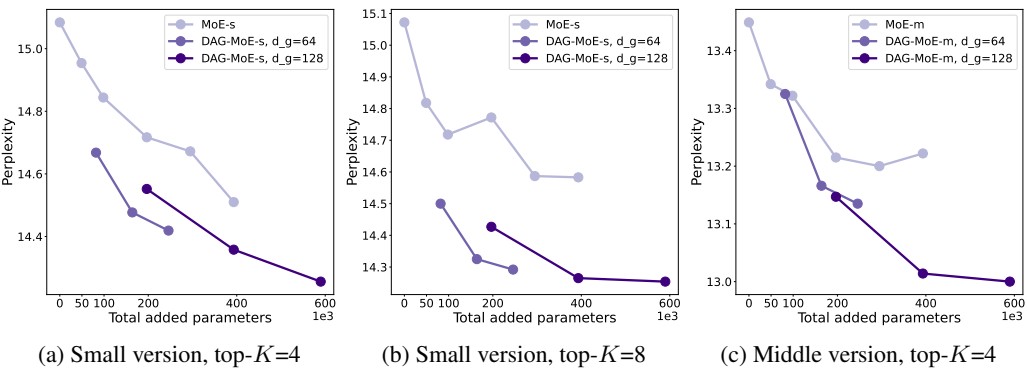

| (a) Small version, top-$K$=4 | (b) Small version, top-$K$=8 | (c) Middle version, top-$K$=4 |

Figure 3: Perplexity of standard MoE and DAG-MoE on Pile subset. The x-axis indicates the additional parameters in the MoE module. For the baseline, it is the size of added shared expert. For DAG-MoE, it is equal to the the number of iteration times parameter size for each iteration.

We first pre-train both DAG-MoE-s, DAG-MoE-m and corresponding standard MoE baseline with 12B tokens. For DAG-MoE, we vary $d_g$ and the depth $L$ with $d_g \in \{64, 128\}$ and $L \in \{1, 2, 3\}$. The main results are shown in Fig. 3, including baseline MoE, DAG-MoE-s, and DAG-MoE-m. For the baseline MoE, we include variants with and without a shared expert; the shared-expert size is encoded on the x-axis as "added parameters," where 0 denotes no shared expert. For DAG-MoE-s and DAG-MoE-m, the x-axis likewise reflecting the added parameters introduced by the DAG learning module. Each curve in the figure corresponds to a fixed $d_g$ while varying $L$ from 1 to 3, and the marker position indicates the total added parameters accumulated over all $L$ iterations.

From Fig. 3, several observations emerge. First, **DAG-MoE consistently achieves lower perplexity than the standard MoE architecture** across nearly all model sizes and settings of $L$ and $d_g$. This indicates that DAG-MoE 's greater expressiveness yields practical performance gains: for a comparable computational budget, the learned DAG structure affords greater flexibility in composing and processing information from the selected experts. More importantly, it is effective under both coarse-grained (top-$K$=4) and fine-grained (top-$K$=8) expert settings, indicating that the DAG learning module applies robustly across MoE configurations for performance gains. Second, with only a small number of additional parameters, DAG-MoE can significantly improve over standard MoE (without shared expert). All these observation directly answer the **Q1**.

To answer **Q2**, we shown the performance improvement over iteration comparing to baseline MoE without shared expert in Fig. 4. First, for both DAG-MoE-s and DAG-MoE-m, given the same $d_g$, performance improves as the number of iterations $L$ increases. In particular, increasing $L$ from 0 to 1 and 1 to 2 brings a substantial gain, For example, for DAG-MoE-s with both top-$K$=4 and top-$K$=8, adding a DAG learning module with $d_g$=64 for a single iteration yields about 0.5 improvement in perplexity. However, the improvement from 2 to 3 is marginal. A plausible explanation is that, for most tokens, one or two iterations suffice to ac-

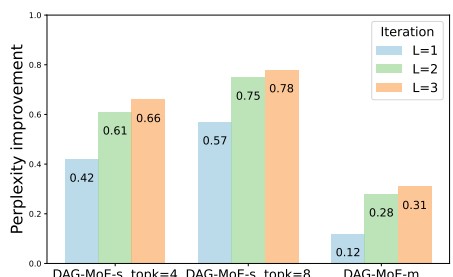

Figure 4: Perplexity improvement for DAG-MoE ($d_g = 64$) over baseline MoE.

curately aggregate information from the selected experts, making additional iterations unnecessary. Second, increasing $d_g$ also helps, but the effect is less pronounced than increasing $L$. For example, in Fig. 3a, DAG-MoE with $d_g = 64$ and $L = 2$ achieves better performance than $d_g = 128$ and $L = 1$, while adding fewer additional parameters. Similar trends are also observed in Fig. 3b.

Finally, to further validate our statement, we pretrain DAG-MoE-l with 40B tokens and evaluate the perplexity on multiple corpus. For DAG-MoE-l, we adopt DAG learning module with $d_g = 256$ and $L = 2$. Corresponding, the baseline model add an shared expert with

Table 1: Pretraining evaluation of DAG-MoE-l

| Perplexity ↓ | Pile | Wiki | FineWeb | C4 |
|---|---|---|---|---|
| MoE-l | 10.51 | 21.08 | 25.38 | 35.21 |
| DAG-MoE-l | **10.27** | **20.54** | **24.69** | **34.21** |

$d_r = 512$. Both DAG-MoE-l and corresponding baseline have 699M parameters. After pretraining, we evaluate the perplexity of the model on both in-domain corpus (Pile) and out-of-domain corpus. The results are shown in Table 1. We can see that DAG-MoE is consistent outperform standard MoE with the same training and parameter budget, especially on the out-of domain corpus. This further validate the superior expressiveness of DAG-MoE and the flexibility brought by the DAG structure. We provide training curve and additional discussion in Appendix C.1.

## 5.3 FINE-TUNING EVALUATION RESULTS

Table 2: Downstream results of DAG-MoE-l

| Accuracy ↑ | PIQA | ARC-e | HellaSwag | GPQA | Lambada | MMLU | BBH | Average |
|---|---|---|---|---|---|---|---|---|
| MoE-l | 47.52 | 24.34 | **25.90** | 21.72 | 8.11 | **24.17** | 16.65 | 24.06 |
| DAG-MoE-l | **50.67** | **25.57** | 25.73 | **27.78** | **11.57** | 24.03 | **17.55** | **26.13** |

To answer **Q3**, we further fine-tune the pretrained DAG-MoE-l and MoE-l on several instruction-tuning dataset. after fine-tuning, we evaluate the model on downstream tasks. Note that both DAG-MoE-l and MoE-l were trained with exactly the same configuration to ensure a fair comparison. The evaluation results are shown in Table 1. We can see that DAG-MoE-l achieve better or similar performance across all downstream task comparing to baseline. This results confirm that the advantage the DAG-MoE obtained can effectively transfer into downstream domain, further confirm the effectiveness of the DAG-MoE.

## 6 CONCLUSIONS AND LIMITATIONS

In this paper, we replace the simple weighted summation in MoE with structural aggregation. By introducing structure and formulating expert aggregation as a DAG, we enlarge the space of expert combinations and enhance model flexibility without modifying the router or expert configurations. To this end, we propose DAG-MoE, which dynamically learns an optimal DAG structure via a lightweight DAG-learning module. Across both language modeling and downstream tasks, DAG-MoE consistently outperforms standard MoE, confirming its effectiveness. However, in the current implementation of DAG-MoE, we impose restrictions on the class of DAGs (number of node at each iteration) that can be learned, which may limit overall expressiveness. Moreover, the problem of identifying the optimal DAG structure for a given token—and how to learn it effectively within the module—remains underexplored. Finally, our evaluation is limited to small-scale training; it is unclear how DAG-MoE would perform at larger scales (e.g., billion-parameter models and trillion-token corpora). We leave these directions to future work.

# 7 REPRODUCIBILITY STATEMENT

We include detailed proof of all theorems provided in the main paper in Appendix A. The detailed implementation, training, and data configuration used to reproduce the results are provided in Appendix B. The code to reproduce the results are provided in anonymous link `https://anonymous.4open.science/r/DAG_MoE-1301/`.

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

# A    DETAILED PROOFS OF ALL THEOREMS

## A.1    DETAILED PROOFS FOR EXPRESSIVENESS OF DAG-MoE

For the completeness, here we first restate the general formula of DAG-style MoE. Now, let $x_i^l$ denote the output of node $v = (l, i)$. For a given DAG $G \in \mathcal{G}(\mathcal{K})$, the corresponding computation in DAG-style MoE can be formulated as:

$$x_i^0 = g_{\mathbf{k}[i]}(x) E_{\mathbf{k}[i]}(x), \quad i = 1, \dots, K, \tag{13}$$

$$x_i^l = AGG(\{x_j^k | (k, j) \in A_i^l\}), \quad i = 1, \dots, n(l), l = 1, \dots, L-1, \tag{14}$$

$$y = AGG(\{x_j^k, | (k, j) \in A_1^L\}). \tag{15}$$

We leverage results from D-VAE (Zhang et al., 2019), which encodes computations over a DAG via an injective aggregation–update scheme executed in topological order. Concretely, for a DAG $G = (\mathcal{V}, \mathcal{A})$ with an initial node $(0, 0)$, the D-VAE encode the DAG via:

$$x_0^0 = \hat{x}_0^0, \tag{16}$$

$$x_i^l = AGG(\{U(\hat{x}_i^l, x_k^j) | (k, j) \in A_i^l\}), \quad i = 1, \dots, n(l), l = 1, \dots, L-1, \tag{17}$$

$$y = AGG(\{x_j^k, | (k, j) \in A_1^L\}). \tag{18}$$

where $AGG$ (aggregation) and $U$ (node update) are injective, and $\hat{x}_i^l$ is the initial feature of node $(l, i)$. In D-VAE (Zhang et al., 2019), the author provide the following conclusion and we restate it here:

**Proposition A.1.** *Let $G$ be a DAG graph with a single initial node $(0, 0)$, the Eq. 16-Eq. 18 can map $G$ to $y$ injectively if $AGG$ and $U$ is injective.*

*Proof.* See Theorem 2 in D-VAE Zhang et al. (2019). □

With the above Proposition, we are ready to prove the Proposition 3.1. We restate it here:

**Proposition A.2.** *Given a top-$K$ experts list $\mathbf{k}$, any DAG-style MoE satisfy Eq. 3-Eq. 5 can injectively encode any $G \in \mathcal{G}(K)$ if $AGG$ is injective.*

*Proof.* To prove the conclusion, we reduce Eq. 13–Eq. 15 to a special case of the D-VAE encoder Eq. 16–Eq. 18. Before reduction, there are several divergence between the definition in DAG-style MoE and D-VAE.

- in D-VAE, the author assumes that any DAG $G$ have a single start node $(0, 0)$. However, here, we assume that we have $K$ start node, each corresponding to an output from one expert. To align it, we notice that all representations are generated by the input token $x$, and $x$ can serve as the node $(0, 0)$ to match the Eq. 16. Then, all previous $K$ start nodes become node at iteration 1, so on so for.

- in D-VAE, each node in $G$ is assumed to have initial feature $\hat{x}_i^l$ and it is integrated through injective $U$ function, while in DAG-style MoE, we do not have it. To align the DAG-style MoE to D-VAE, we assume that each node in $G$ will have same $\mathbf{0}$ initialized input feature except node $(0, 0)$. Note that it is obvious that these modification will not affect the expressiveness of DAG-style MoE as we do not have initial features.

Given all these change, we can rewrite the formulation of DAG-style MoE as follows:

$$x_0^0 = x, \tag{19}$$

$$x_i^1 = g_{\mathbf{k}[i]}(x_0^0) E_{\mathbf{k}[i]}(U(\mathbf{0}, x_0^0)), \quad i = 1, \dots, K, \tag{20}$$

$$x_i^l = AGG(\{U(\mathbf{0}, x_j^k) | (k, j) \in A_i^l\}), \quad i = 1, \dots, n(l), l = 2, \dots, L, \tag{21}$$

$$y = AGG(\{x_j^k, |(k, j) \in A_1^{L+1}\}). \tag{22}$$

We observe that the above equations almost coincide with the D-VAE encoder, except several components. First, we add $U$ to the DAG-style MoE to match the equation in D-VAE. However, as the initial feature here is $\mathbf{0}$, it will not affect the expressiveness of the model, and we can actually combine it with $AGG$ as new aggregation function. Second, at the first iteration, under our DAG definition, every node at iteration 1 has exactly one incoming predecessor, using $E(x)$ as the $AGG$ function is equivalent to original formulation, as each node aggregates from a singleton parent. Meanwhile, the $g_i$ and $E_i$ can be easily implemented as injective function. Consequently, the Eq. 20 can be further rewrites as:

$$x_i^1 = AGG(\{U(\mathbf{0}, x_0^0)\}), \quad i = 1\ldots, K. \tag{23}$$

Now, the process of DAG-style MoE can perfectly match with D-VAE and the theorem is directly proved given Proposition A.1. We omit details here. □

Next, we prove the Theorem 3.2. We restate it here:

**Theorem A.3.** *Given a top-$K$ experts set $\mathbf{k}$, any DAG-style MoE satisfy Eq. 13-Eq. 15 is strictly more powerful than standard MoE with Eq. 1 if $AGG$ is injective.*

*Proof.* First, it is straightforward that DAG-style MoE is at least as powerful as standard MoE. Set $G$ to have $K$ isolated nodes (no interactions among the top-$K$ experts; see the left panel of Fig. 1) and take $L = 0$. Then Eq. 14 vanishes and Eq. 15 reduces to a readout over $\{x_i^0 = g_i(x)E_i(x)\}_{i=1}^K$. With AGG chosen as summation, this exactly recovers the standard MoE weighted sum.

Next, we show that DAG-style MoE can map different DAG structures to different outputs, whereas standard MoE always produces the same representation for a fixed set of selected experts. Consider the tree in the middle of Fig. 1 and define two DAGs that differ only in how two leaves are paired at the first depth:

$$G_1 = (\mathcal{V}_1, \mathcal{A}_1), \quad \mathcal{A}_1 = \{A_1^1 = \{(0, 1), (0, 2)\}, \ A_2^1 = \{(0, 3), (0, 4)\}, \ A_1^2 = \{(1, 1), (1, 2)\}\},$$

$$G_1 = (\mathcal{V}_1, \mathcal{A}_1), \quad \mathcal{A}_1 = \{A_1^1 = \{(0, 1), (0, 3)\}, \ A_2^1 = \{(0, 2), (0, 4)\}, \ A_1^2 = \{(1, 1), (1, 2)\}\},$$

Let $x_{g,i}^0$ denote the initial representation of node $(0, i)$ in graph $G_g$. Assume the four leaf representations are the same in both graphs, i.e., $x_{1,i}^0 = x_{2,i}^0$ for $i = 1, \ldots, 4$, and that $x_{\cdot,2}^0 \neq x_{\cdot,3}^0$. Then the predecessor multisets at depth 1 differ between $G_1$ and $G_2$ (one pairs $(0, 1)$ with $(0, 2)$, the other pairs $(0, 1)$ with $(0, 3)$), hence the depth-1 node states differ because AGG is injective on multisets. By composition of injective maps, all downstream states—and therefore the final outputs—also differ. Formally, by Proposition A.2, DAG-style MoE maps $G_1$ and $G_2$ to distinct outputs. In contrast, standard MoE aggregates the same leaf set $\{x_{\cdot,i}^0\}_{i=1}^4$ by a permutation-invariant weighted sum, yielding identical outputs for $G_1$ and $G_2$. Thus, DAG-style MoE strictly separates these two structures while standard MoE does not, which conclude the proof. □

### A.2 DETAILED DISCUSSION ON DYNAMIC PROGRAMMING.

#### A.2.1 DEFINITION OF DYNAMIC PROGRAMMING.

In this section, we formally define the dynamic programming problem based on previous work Feng et al. (2023). Formally, a general DP algorithm can be characterized via three key ingredients: state space $\mathcal{I}$, transition function $T$, and aggregation function $AGG$. Given a DP problem with $N$ input sequences $\boldsymbol{s}^{(1)}, \cdots, \boldsymbol{s}^{(N)}$, denote the problem size to be the vector $\boldsymbol{n} = (|\boldsymbol{s}^{(1)}|, \cdots, |\boldsymbol{s}^{(N)}|)$. Given the fixed problem size $\boldsymbol{n}$, there is an associated states space $\mathcal{I}_{\boldsymbol{n}} \subset \mathcal{I}$ representing the finite set of decomposed subproblems, where each state $i \in \mathcal{I}_{\boldsymbol{n}}$ is an index signifying a specific subproblem. The size of the state space $\mathcal{I}_{\boldsymbol{n}}$ grows with the problem size $\boldsymbol{n}$. We denote by $dp(i)$ the answer along with other information about the DP process of subproblem $i$. Furthermore, there is a partial order relation between different states: we say state $j$ precedes state $i$ (denoted as $j \prec i$) if subproblem $j$ should be solved before subproblem $i$. This partial order naturally creates a DAG within the state space, thereby establishing a reasoning chain that can be approximated by the DAG-style MoE.

In the paper, we focus on a restricted setting where each state $i$ only depends on (i) a finite number of tokens in the input sequence $\boldsymbol{s}$ and (ii) a finite number of previous states. Under this assumption, the transition function $T$ can be generally written as:

$$
\begin{aligned}
dp(i) &= f(\boldsymbol{n}, i, s_{\boldsymbol{g}(\boldsymbol{n},i)}, dp(\boldsymbol{h}(\boldsymbol{n},i))) \\
&= f(\boldsymbol{n}, i, s_{g_1(\boldsymbol{n},i)}, \cdots, s_{g_J(\boldsymbol{n},i)}, dp(h_1(\boldsymbol{n},i)), \cdots, dp(h_P(\boldsymbol{n},i))),
\end{aligned}
\tag{24}
$$

where functions $f, \boldsymbol{g}, \boldsymbol{h}$ fully determine the transition function $T$ and have the following form $f : \mathbb{N}^N \times \mathcal{I} \times \mathcal{X}^J \times \mathcal{Y}$, $\boldsymbol{g} : \mathbb{N}^N \times \mathcal{I} \to (\mathbb{N} \cup \{\emptyset\})^J$, $\boldsymbol{h} : \mathbb{N}^N \times \mathcal{I} \to (\mathcal{I} \cup \{\emptyset\})^P$. Here, the state space $\mathcal{I}$, input space $\mathcal{X}$, and DP output space $\mathcal{Y}$ can be arbitrary domains, and $J, P$ are constant integers. If state $i$ depends on less than $J$ input tokens or less than $P$ previous states, we use the special symbol $\emptyset$ to denote a placeholder, such that all terms $s_\emptyset$ and $dp(\emptyset)$ are unused in function $f$. To further explain, $\boldsymbol{g}(\boldsymbol{n}, i)$ indicates the input index that that is used for compute transition function $dp(i)$, while $\boldsymbol{h}(\boldsymbol{n}, i)$ indicates all previous DP states that is required. After solving all subproblems, the aggregation function $AGG$ is used to combine all results and obtain the final answer. We consider a general class of aggregation functions with the following form:

$$
AGG(\{(i, dp(i)) : i \in \mathcal{I}_{\boldsymbol{n}}\}) = u(\square_{i \in \mathcal{A}_{\boldsymbol{n}}} dp(i)),
\tag{25}
$$

where $\mathcal{A}_{\boldsymbol{n}} \subset \mathcal{I}_{\boldsymbol{n}}$ is a set of states that need to be aggregated, $\square$ is an aggregation function such as $min$, $max$, or $\sum$, and $u : \mathcal{Y} \to \mathcal{Z}$ is any function where $\mathcal{Z}$ denotes the space of possible answers. Now, we describe how to construct the DAG based on a DP problem.

**Definition A.4.** Given a DP problem with input sequence $\boldsymbol{s}^{(1)}, \cdots, \boldsymbol{s}^{(N)}$, problem size $\boldsymbol{n} = (|\boldsymbol{s}^{(1)}|, \cdots, |\boldsymbol{s}^{(N)}|)$, and transition function specified in Eq. 24, we define the DAG $G_{dp}$ formulated by the solving process of DP as follows:

- There are $n$ node at depth 0, while each node store the required input $s_i$ used for solving subproblem $i \in n$.

- for depth $l > 0$ , Each node $(l, i)$ represents a solution $dp(i)$ that satisfying for any $j \in \boldsymbol{h}(\boldsymbol{n}, i)$, the depth of $dp(j)$ is less than $l$.

- for the last depth $L(dp)$, it contains a single node represents the output $y$ of DP problem.

- The Adjacency list $\mathcal{A}$ is defined by $A_i^l = \{(d(j), j) | j \in \boldsymbol{h}(\boldsymbol{n}, i)\} \cup \{(0, j) | j \in \boldsymbol{g}(\boldsymbol{n}, i)\}$ for $l = 1, \ldots, L - 1$, where $d(j)$ is the depth of $dp(j)$.

- Finally, we have $A_0^{L(dp)} = \{(d(i), i) | i \in \mathcal{A}_{\boldsymbol{n}}\}$.

There is two elements worth further discussion. First, the $d(i)$ define the depth of subproblem $i$ in the DAG, which should be the smallest iteration that the $dp(i)$ can be obtained. In another word, the $d(i)$ is the smallest iteration while all $dp(j), j \in \boldsymbol{h}(\boldsymbol{n}, i)$ are ready. $L(dp)$ is the depth of DAG $G$, which is the smallest iteration to obtain the final answer of DP problem.

### A.2.2 Longest increasing subsequence problem and an example on how DAG-MoE can simulate its solving process.

In this section, we describe one representative DP problem: the longest increasing subsequence(LIS) problem. The LIS problem aims to compute the length of the longest increasing subsequence given an input sequence $\mathbf{s} \in \mathbb{N}^n$. Define $\tilde{\mathbf{s}}$ as a subsequence of $\mathbf{s}$ if there exists indices $1 \leq i_i \leq \cdots \leq i_{|\tilde{\mathbf{s}}|} \leq n$ such that $\tilde{s}_k = s_{i_k}$ holds for all $k \in 1, \ldots, |\tilde{\mathbf{s}}|$. A sequence $\tilde{\mathbf{s}}$ is called increasing if $\tilde{s}_1 \leq \cdots \leq s_{|\tilde{\mathbf{s}}|}$. The LIS problem aims to find an increasing subsequence of $\mathbf{s}$ with maximal length. Let $h(n, i) = \{j | s_j < s_i\}$ be the predecessor index set include all element index $j$ that satisfy $s_j < s_i$, the transition function of LIS problem can be written as:

$$
dp(i) = 1 + \max(\{dp(j) | j \in h(n, i)\}), \quad \max(\emptyset) = 0.
\tag{26}
$$

The final solution can be obtained by:

$$
y = \max_i(\{dp(i) | i = 1, \ldots, n\})
\tag{27}
$$

Here we show a small example with a sequence $[3, 1, 2, 4]$ in top of Fig. 5. By iteratively performing the transition function, we can obtain the final answer 3, as described in the left of Fig. 5. At the

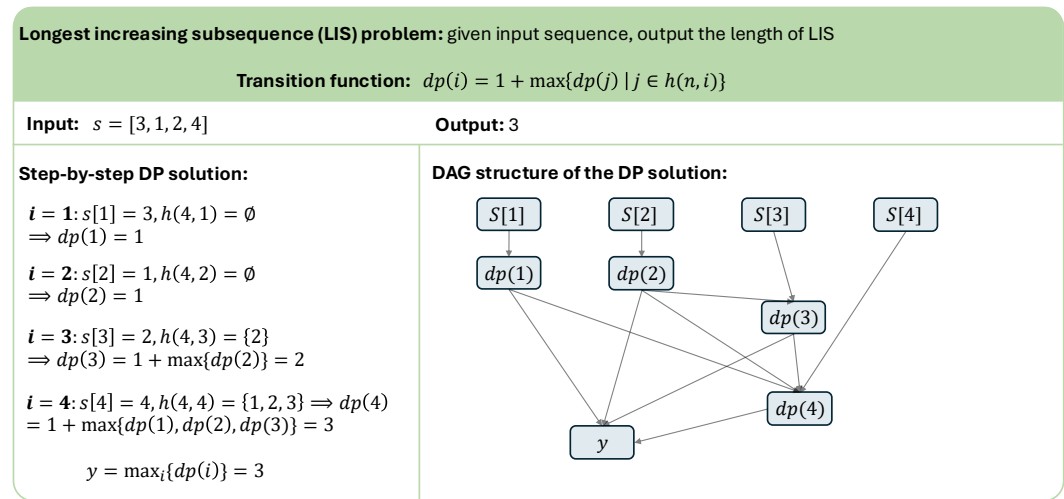

Figure 5: An example of LIS problem and corresponding DAG structure

same time, the process can be convert into a DAG model specify the computation, as shown in the right part of Fig. 5.

Now, suppose we have a DAG-style MoE which choose top-4 experts and the DAG structure in the MoE block are exactly the same as the $G(dp)$ of the LIS problem shown in Fig. 5. Then, if we input the sequence into the model as ask the model to predict the answer at the next token $x$. In the first attention block, Given an well-trained strong attention module, we can easily retrieve all sequence into the $x$ to produce representation $\hat{x}$. Then, at the MoE block, given the universal approximation ability of the MLP, it is also easy to let each expert to maintain only one element in the sequence, which corresponding to the first depth of the graph $G(dp)$ in Fig. 5. Next, we aggregate the expert output given the $G(dp)$ to produce the final answer. It is obvious that given powerful $AGG$ implementation and sufficient training, each $AGG$ can approximate the mapping from input to the solution of $dp(i)$ and the final $AGG$ can approximate the final answer based on each $dp(i)$. Thus, it is theoretical possible to directly simulate the overall solving process of the LIS example in Fig. 5 with single DAG-style MoE layer.

### A.3 PROOF OF THEOREM 3.3

Given all the definitions above, we are ready to prove the Theorem 3.3. The theorem is largely based on the proof of Theorem 4.7 in (Feng et al., 2023). Thus, the proof will also under the Assumption 4.2-4.5 in (Feng et al., 2023), we omit it here and refer reader to the original paper for detailed assumption. We restate Theorem 3.3 in the below:

**Theorem A.5.** *For any integer $n \in \mathbb{N}$, Consider any DP problem satisfying Assumption 4.2 to 4.5 in (Feng et al., 2023) with corresponding DAG $G(dp)$ and problem size length less than $O(K log(n))$, there exists a log-precision constant depth transformer composed of (i) multi-head attention block and (ii) at least one DAG-MoE block with top-$K$ experts and number of iteration $L \geq L(dp)$, with hidden dimension $d$ and parameter size bounded by $O(poly(n))$ that can output the correct DP answer.*

*Proof.* Before proving, note that there are several major difference between the Theorem A.5 and Theorem 4.7 in (Feng et al., 2023). First, Theorem 4.7 focus on DP problem with size $n$ and prove the transformer can solve it step by step with Chain-of-Though mechanism. The key here is that, by iteratively execute the transformer, we can obtain $O(n)$ length sequence to store all input states and intermediate states, and then leverage the attention to pick the information used for the next step. In Theorem A.5, what we want to show is that within a single DAG-style MoE block, we can simulate the DP solving process with DAG. However, one limitation is that we only have $K$ experts, which is obtained from single input token. Given the log-precision limitation (See (Feng et al., 2023) for detailed definition), it can only store $log(n)$ amount of information, which is not possible to store input with length of $O(n)$. Therefore, we reduce the problem size to $O(K log(n))$ so that we can

allow single representation to store all required information, like problem input. Next, we show the proof sketch on how to construct DAG-MoE layer to implement the DAG of DP solving process.

**Input format**, we consider represent DP problem by a sequence of tokens:

$$\boldsymbol{s}^{(1)} \quad | \quad \cdots \quad | \quad \boldsymbol{s}^{(N)} \quad | \quad \text{final answer}$$

**Block 1.** The first block contains several layers it function as simple COPY machine (see Lemma C.7. in (Feng et al., 2023) for feasibility proof). Specifically, we use the attention module to select and copy the first all $O(K log(n))$ input sequence into the output token representation, with the each slice of the representation store one input element. Given results from Feng et al. (2023), this can be performed using attention and MLP.

**Block 2.** Once the token representation contains all input information, we use a single DAG-style MoE layer to perform the DP process. Specifically, we let the structure of DAG in DAG-style MoE to be exactly the same as $G(dp)$ (which can be learned through carefully designed module or predefined). Next, given a input token representation that store all input sequence, each top-$K$ expert is used to copy $1/K$ part of information from the input. This can be implement by set 1 on the corresponding slice and 0 for other slice in the MLP weight. This formed the node at depth 0. Note that in the Definition A.4, depth 0 has $n$ node and each node only store 1 input. However, here we take the advantage of the hidden representation and let each node to store $log(n)$ input. Then, at each depth, we simply perform the $AGG$ to compute the solution of each subproblem $dp(i)$ using the following DAG-style computation:

$$x_i^l = AGG(x_j^k | (k, j) \in A_i^l), \tag{28}$$

where $x_i^l$ represent the solution of subproblem $dp(i)$. Since here we define the $AGG$ as an injective function, we can implement it by sum/min/max + MLP, while the sum/min/max is used to aggregate result from previous subproblem, while the MLP is used to compute the result for the current problem. Given the assumption 4.2 to 4.5 in (Feng et al., 2023), it is sufficient for $AGG$ to learn such mapping from input to answer of subproblem $dp(i)$. At the final depth $L(dp)$, we perform the following computation:

$$y = AGG\{dp(i) | i \in \mathcal{A_n}\}, \tag{29}$$

where $AGG$ here can be implemented by $u + \square$, which is typically MLP + sum/min/max. It is easy to obvious that we can learn such mapping by well implemented $AGG$. We omit the detailed implementation of each step here and refer reader to (Feng et al., 2023) example implementation.

□

## B  MORE DETAILS ON THE MODEL IMPLEMENTATION AND TRAINING

In this section, we provide more details on the model implementation and training.

### B.1  PRETRAINING

We summarize the detailed configuration and hyperparameters for both DAG-MoE and baseline in Table 3 and discuss details below.

**Model configuration.** For both the baseline MoE and DAG-MoE, we implement the model on top of Llama3.1-8B (Dubey et al., 2024). We retain the tokenizer, vocabulary table, attention module, and FFN design of Llama3.1-8B, but reduce the number of Transformer layers and the hidden dimension size due to resource constraints. The MoE module follows the standard token-choice router from Switch Transformer (Fedus et al., 2022) with balance loss, except that expert scores are computed with a Sigmoid function instead of SoftMax, as we find Sigmoid performs better in practice. In addition, we apply router Z-loss to regularize the logits, following prior work in state-of-the-art MoE models (Muennighoff et al., 2025; Zoph et al., 2022; Tian et al., 2025a). For DAG-MoE, we incorporate the DAG learning module on top of the standard MoE block, varying both the graph dimension $d_g$ and the depth $L$ during evaluation. For the baseline MoE, we include a shared expert to ensure parameter parity with DAG-MoE. The shared expert adopts the same architecture as other experts, and its hidden dimension $d_r$ is adjusted to match the additional parameters introduced by the DAG learning module. Our implementation is based on the Hugging Face Transformers library (Wolf et al., 2019).

Table 3: The model configuration and hyper-parameter setting for DAG-MoE and baseline

| configuration | DAG-MoE-s & MoE-s | DAG-MoE-m & MoE-m | DAG-MoE-l & MoE-l |
|---|---|---|---|
| model hidden size $d$ | 512 | 512 | 1024 |
| number of layer | 4 | 6 | 8 |
| number of attention heads | 32 | 32 | 32 |
| number of key-value heads | 8 | 8 | 8 |
| number of experts $N$ | 32/64 | 32 | 32 |
| expert hidden size $d_r$ | 256/128 | 256 | 512 |
| balance loss coefficient | 0.01 | 0.01 | 0.01 |
| Router Z loss coefficient | 0.001 | 0.001 | 0.001 |
| dropout ratio | 0.0 | 0.0 | 0.0 |
| optimizer | adamW | adamW | adamW |
| adam $\beta_1$ | 0.9 | 0.9 | 0.9 |
| adam $\beta_2$ | 0.999 | 0.999 | 0.999 |
| adam $\epsilon$ | 1e-8 | 1e-8 | 1e-8 |
| weight decay | 0.1 | 0.1 | 0.1 |
| learning rate | 5e-4 | 5e-4 | 3e-4 |
| warmup steps | 2000 | 2000 | 2000 |
| decay ratios | 0.2 | 0.2 | 0.2 |
| learning rate scheduler | WSD | WSD | WSD |
| batch size | 256 | 256 | 512 |

Table 4: The statisic of pre-training data

| Pile | # sample | # train token | # val. token |
|---|---|---|---|
| 12B | 10,000,000 | 12,499,025,920 | 1,388,838,912 |
| 40B | - | 39,321,600,000 | - |

**Data.** We use Pile (Gao et al., 2020) as our pretraining corpus, which is a large-scale open source lauange pretraining dataset. Specifically, we conduct two set of pretraining. For DAG-MoE-s and DAG-MoE-m, we randomly sample a subset from Pile with 10 million samples and use 10% as the evaluation set, which results in about 12B tokens for training and 1.3B for evaluation. For DAG-MoE-l, we using data streaming to train the model on the Pile dataset with 37500 steps, results in about 40B training tokens. The detailed data statistics can be found in Table 4. For pretraining, we split the original sample into sub-samples with each sample of a sequence length of 2048. We further include out-of-domain corpus for additional evaluation on DAG-MoE-l. Specifically, we add FineWeb-Edu (Lozhkov et al., 2024), Wikipedia text (Thrush et al., 2022), and C4 (Raffel et al., 2020). For each dataset, we randomly sample 500,000 samples as evaluation corpus.

**Training.** We train all models from scratch on the pretraining dataset. Optimization is performed using AdamW with default settings, and the maximum learning rate is adjusted according to the model size. Following prior works (Bae et al., 2025; Hu et al., 2024; Tian et al., 2025a), we employ the WSD (warmup–stable–decay) scheduler, which not only improves convergence but also enables checkpoint reuse during training. The warmup phase is fixed to 2,000 steps, and the decay ratio is set to 20%. Training is conducted on the causal language modeling task with cross-entropy loss. The coefficients for balance loss and Z-loss are set to $0.01$ and $0.001$, respectively. We apply a weight decay of $0.1$, and the global batch size is varied across models. All experiments are run with DeepSpeed ZeRO-2 (Rajbhandari et al., 2020) on 8 NVIDIA A100 GPUs. Our implementation builds on LlamaFactory (Zheng et al., 2024).

## B.2 FINE-TUNING AND DOWNSTREAM EVALUATION

**Model Configuration.** For fine-tuning, we directly use the pretrained DAG-MoE-l and MoE-l as the based model. For DAG-MoE-l, we set $d_g = 256$ and $L = 2$ in the DAG learning module. Correspondingly, for MoE-l, we add a shared expert with hidden size 512, results in both model have 699M parameters. See Table 3 for the detailed model configuration.

**Data.** For fine-tuning, we incorporate multiple datasets, including Alpaca (Taori et al., 2023), Open-Platypus (Lee et al., 2023), SlimOrca (Mukherjee et al., 2023), MathInstruct (Yue et al., 2023),

Open-r1-math, and MetaMathQA (Yu et al., 2023). We didn't do any down/up sampling for each dataset. All datasets are obtained from Huggingface.

**Training.** We train the model with 3 epoch on all the data with constant learning rate $2e - 5$. The coefficient for balance loss and Z-loss are the same as pretraining, while we set the weight decay to 0 to allow model to adjust its behavior given fine-tuning tasks. For fine-tuning, we set batch size to 256, which results in total 15435 steps for 3 epoch. All experiments are run with DeepSpeed ZeRO-2 Rajbhandari et al. (2020) on 8 NVIDIA A100 GPUs. Our implementation builds on LlamaFactory Zheng et al. (2024).

**Evaluation.** After fine-tuning, we evaluate both the DAG-MoE-l and MoE-l on downstream tasks including PIQA (Bisk et al., 2020), ARC-e (Clark et al., 2018), HellaSwag (Zellers et al., 2019), GPQA (Rein et al., 2024), Lambada (Paperno et al., 2016), MMLU (Hendrycks et al., 2009), and BBH (Suzgun et al., 2022). PIQA (Bisk et al., 2020) tests physical commonsense by asking models to choose the more plausible solution to everyday tasks. ARC-e (Clark et al., 2018) is the "Easy" subset of the AI2 Reasoning Challenge, consisting of grade-school science multiple-choice questions. HellaSwag (Zellers et al., 2019) evaluates grounded commonsense by requiring selection of a plausible continuation for a short scenario. GPQA (Rein et al., 2024) measures expert-level knowledge and reasoning with graduate-difficulty multiple-choice questions. Lambada (Paperno et al., 2016) assesses broad-context language modeling via last-word prediction that requires understanding a long passage. MMLU (Hendrycks et al., 2009) benchmarks multi-task language understanding across 57 academic subjects in a few-shot multiple-choice format. BBH (Suzgun et al., 2022) (Big-Bench Hard) comprises 23 challenging reasoning tasks probing compositionality, logic, and algorithmic generalization. For HellaSwag, we use 10-shot setting; For MMLU, we use 5-shot setting; For all other benchmarks, we adopt 0-shot setting without CoT. The evaluation is conducted through Opencompass (Contributors, 2023).

## C   ADDITIONAL DISCUSSION ON THE EXPERIMENTS

### C.1   PRETRAINING OF DAG-MoE-L

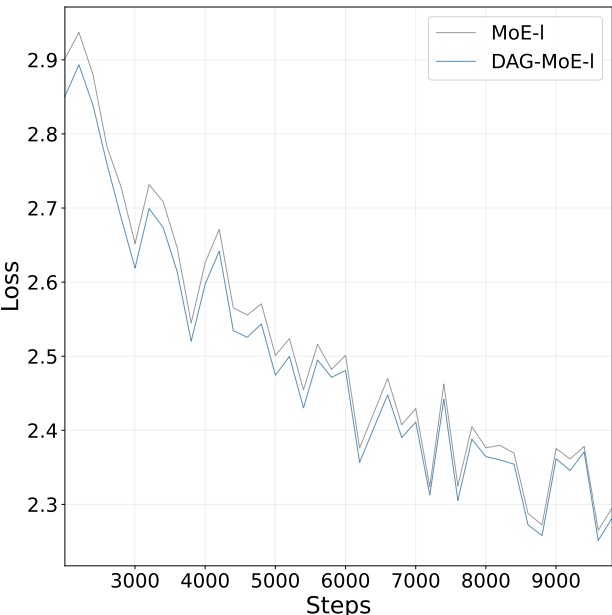

Figure 6: Pretraining loss curve of DAG-MoE-l and MoE-l

In this section, we provide additional discussion on the pretraining of DAG-MoE-l. Specifically, we present the pretraining loss curves for both DAG-MoE-l and MoE-l from 1,000 to 10,000 steps in Fig. 6. We observe that DAG-MoE-l exhibits substantially faster early-stage convergence, evidenced by a large loss gap at the beginning of training, which supports the claim that DAG-MoE

offers greater flexibility than standard MoE. As training progresses, however, the gap narrows and stabilizes toward the end. We hypothesize that this occurs because both models are relatively small and reach similar optima at this data scale. We plan to evaluate DAG-MoE at larger model sizes and with larger training corpora in future work.

## D    LLM USAGE

We used large language models solely for writing assistance and polishing (grammar, clarity, and style). All technical ideas, experiments, and analyses were designed and executed by the authors.

