# OpenReview forum: "DAG-MoE: From Simple Mixture to Structural Aggregation in Mixture-of-Experts"
_ICLR.cc/2026/Conference — Submitted to ICLR 2026_

### Official Review · Reviewer_wjrd · 2025-10-30

**Soundness:** 3
**Presentation:** 2
**Contribution:** 3
**Rating:** 4
**Confidence:** 4

**Summary:**

This paper proposes **DAG-MoE**, a Mixture-of-Experts framework that replaces the conventional weighted-sum aggregation of expert outputs with a **Directed Acyclic Graph (DAG)-based structural aggregation**.  The method introduces a lightweight DAG-learning module that learns inter-expert dependencies among the top-$K$ activated experts, enabling multi-step reasoning and richer composition. Experiments on scaled LLaMA3 models (12B–40B tokens) show consistent perplexity reductions and downstream task improvements over standard MoE baselines, under both coarse- and fine-grained expert settings.  DAG-MoE enhances expressiveness without modifying the router or increasing computational cost, suggesting a promising new direction for scaling sparse architectures.

**Strengths:**

- **Originality:** The paper proposes a novel perspective on Mixture-of-Experts (MoE) scaling by shifting focus from routing optimization to **structural aggregation**. The introduction of a DAG-based aggregation mechanism is an original and conceptually elegant idea that broadens the expressiveness of sparse expert models.
- **Technical Quality:** The method is well-motivated and mathematically consistent. The DAG-learning module is lightweight and integrates smoothly with existing MoE architectures without adding significant computation.
- **Empirical Results:** Extensive experiments on both coarse- and fine-grained settings (12B–40B tokens) demonstrate consistent gains in perplexity and downstream tasks, supporting the claim that DAG-MoE improves compositional reasoning.
- **Clarity and Impact:** The paper is generally clear, well-structured, and provides a new direction for improving MoE efficiency and flexibility, which could inspire follow-up research on structured sparsity and inter-expert reasoning.

**Weaknesses:**

1. **Incomplete Theoretical Foundation:**
   While Theorems 3.1–3.3 claim that DAG-MoE possesses strictly greater expressiveness than standard MoE, the proofs are relegated to the appendix with only high-level intuition in the main text. The mapping between DAG-MoE and message-passing networks (Xu et al., 2018) is conceptually appealing but not quantitatively analyzed. It remains unclear how the injectivity of `AGG` translates into measurable capacity or reasoning depth. A formal bound or representational comparison (e.g., VC dimension or Lipschitz analysis) would make the argument convincing.

2. **Overly Constrained Structural Design:**
   The DAG learning module fixes both the number of nodes per layer (`n(l)=K`) and layer-wise connectivity (`l−1→l`), effectively reducing the search space to a shallow subset of possible DAGs. This design undermines the claimed “exponential expert-combination space.” It would be more convincing to allow adaptive node counts or cross-layer connections and compare their learned structures.

3. **Ambiguous Learning Dynamics and Stability:**
   The module introduces multiple projection matrices (`W_down`, `W_edge`, `W_node`, `W_up`) and relies on zero-weight initialization for `W_up`. However, no ablation is provided to separate their effects or to examine convergence stability. Since the DAG structure evolves across iterations, training dynamics could oscillate or collapse, yet the paper does not report variance or gradient stability metrics.

4. **Weak Empirical Validation of Efficiency:**
   The FLOPs analysis in §3.3 is simplified and omits the cost of sequential multi-iteration computation. Although authors claim the module is “lightweight,” no actual runtime, latency, or GPU utilization comparisons are given. The efficiency claim would require profiling against strong baselines such as DeepSpeed-MoE or GLaM to demonstrate real-world scalability.

5. **Limited Experimental Scope:**
   Experiments are confined to reduced-scale LLaMA3 variants (≤8 layers, ≤1B parameters), which are far smaller than standard large-scale MoE setups. It remains uncertain whether DAG-MoE scales effectively to 7B–70B models or retains stability under mixed precision training. Moreover, downstream tasks are mostly text-based; multimodal or multilingual tasks would strengthen the generality claims.

6. **Ablation and Hyperparameter Analysis Insufficient:**
   The impact of key hyperparameters (`L`, `dg`, `K`) is partially shown in Fig. 3–4, but no quantitative table summarizes sensitivity or trade-offs. For instance, increasing `L` improves perplexity but may also raise compute cost; this balance is not analyzed. Likewise, there is no investigation into how routing entropy or balance loss interacts with the learned DAG structures.

7. **Missing Comparison to Contemporary MoE Systems:**
   The baselines are limited to standard token-choice MoE and a “shared expert” variant. There is no direct comparison with recent structured or hierarchical MoEs (e.g., BASE Layers, Router-XL, Expert-Choice MoE, or MoEfication). Without these, it is hard to attribute performance gain specifically to DAG-based aggregation rather than general parameter increase.

**Questions:**

> I encourage the authors to thoroughly address the weaknesses and questions raised in this review. If the authors can provide detailed explanations and in-depth clarifications during the rebuttal, and if the revised version demonstrates substantial progress in both clarity and improvement, I will be willing to reassess the manuscript and **adjust my overall rating** accordingly, based on the quality and depth of the revision.

---

1. **On Theoretical Justification:**
   The paper claims that DAG-MoE is strictly more expressive than standard MoE (Theorems 3.1–3.3), but only intuitive reasoning is given in the main text. Could the authors provide a more formal quantitative analysis (e.g., capacity bounds, Lipschitz continuity, or representational diversity) that supports this claim? Additionally, how does the injectivity of `AGG` practically translate to measurable improvements in reasoning or compositionality?

2. **On Structural Constraints:**
   The DAG learning module fixes `n(l)=K` and restricts edges to `l−1→l` connections. Have the authors tested more flexible structures, such as adaptive node counts or cross-layer edges? Would these lead to higher expressiveness or better empirical performance? Please discuss the trade-offs behind this design choice.

3. **On Learning Stability:**
   Since the DAG module iteratively updates connections and projections, how stable is training across seeds or epochs? Did the authors observe oscillations, vanishing gradients, or unstable convergence when increasing the number of iterations `L`? Any evidence such as gradient norms or training variance would be helpful.

4. **On Efficiency and Overhead:**
   The FLOPs analysis in §3.3 does not include the sequential cost of multiple DAG iterations. Could the authors provide empirical runtime comparisons (e.g., wall-clock time, latency, GPU utilization) against strong baselines like DeepSpeed-MoE or GLaM to support the “lightweight” claim?

5. **On Experimental Scale and Generalization:**
   All experiments are performed on small-scale LLaMA3 variants (<1B parameters). Have the authors tested DAG-MoE at larger scales (e.g., 7B–70B models) or in multilingual/multimodal settings? How confident are the authors that the observed improvements will persist when scaling up?

6. **On Hyperparameter Sensitivity:**
   Figures 3–4 illustrate the effects of `L` and `dg`, but there is no quantitative table showing sensitivity or compute trade-offs. Could the authors report how perplexity and compute cost change as `L`, `dg`, and `K` vary? This would clarify whether DAG-MoE’s gains come from structure learning or simply more parameters.

7. **On Comparison with Other MoE Frameworks:**
   Why were recent hierarchical or structured MoE baselines (e.g., BASE Layers, Router-XL, Expert-Choice MoE, MoEfication) not included in the comparison? Since these methods also address routing or structural efficiency, including them would help isolate the novelty and contribution of DAG-based aggregation.

---

### Official Review · Reviewer_89Ch · 2025-10-31

**Soundness:** 3
**Presentation:** 3
**Contribution:** 3
**Rating:** 6
**Confidence:** 4

**Summary:**

This paper proposes DAG-MoE to fix traditional MoE’s problems. Traditional MoE uses permutation-invariant weighted summation for expert aggregation, which limits flexibility. It also has routing overhead with fine-grained experts. DAG-MoE replaces summation with DAG-based aggregation via a lightweight module. This module learns optimal DAGs for selected experts. DAG-MoE improves expressiveness and enables multi-step reasoning. It outperforms traditional MoE in pretraining on the Pile dataset and fine-tuning on tasks like PIQA and MMLU.

**Strengths:**

It innovatively uses DAG-based aggregation to replace traditional MoE’s weighted summation, solving the flexibility limitation of MoE and enabling multi-step reasoning (e.g., simulating dynamic programming).

The DAG learning module is lightweight with low overhead, and it is compatible with existing MoE components (no need to modify experts or routers), making it easy to apply to LLMs.

**Weaknesses:**

There is no analysis of the DAG learning dynamics, such as which structural patterns the module learns for different tasks.

The paper lacks interpretability in terms of reasoning: there is no discussion on how the learned DAG structure influences the model’s task performance.

There are no critical ablation studies to validate key DAG components, which makes it unclear what drives the model’s performance.

**Questions:**

No further questions, see above.

---

### Official Review · Reviewer_vE6c · 2025-10-31

**Soundness:** 3
**Presentation:** 3
**Contribution:** 2
**Rating:** 2
**Confidence:** 4

**Summary:**

The paper proposes DAG-MoE, which replaces the standard weighted-sum aggregation in Mixture-of-Experts with a structural aggregation mechanism based on a directed acyclic graph. This design expands the expert-combination space and enables potential multi-step reasoning within a single MoE layer. A lightweight DAG-learning module is introduced to automatically learn the optimal aggregation structure among selected experts.

**Strengths:**

The paper introduces a structured aggregation mechanism (DAG-style) for MoE, providing an alternative to weighted summation. The formulation is clear and theoretically grounded.

**Weaknesses:**

(1) The proposed DAG aggregation is conceptually similar to existing message-passing or recursive expert mechanisms (e.g., S′MoRE, Chain-of-Experts).
(2) The theoretical analysis is largely derived from prior work (Deep Sets, D-VAE) without new insight.
(3) Experiments are limited to small-scale settings, lacking comparisons to recent MoE variants and large-model validation.

**Questions:**

The proposed DAG-MoE seems conceptually similar to prior structured or recursive MoE models. Could the authors clarify the key novelty and advantages over existing approaches such as S′MoRE or Chain-of-Experts?
How interpretable or consistent are the learned DAG structures across inputs or training runs?
Can the method scale effectively to large-scale MoE/LLM settings beyond small models?
Would stronger baselines or ablations (e.g., fixed DAGs, more comparisons) help isolate the contribution of the DAG-learning module?
Could the authors provide clearer evidence or intuition for the claimed multi-step reasoning ability?

---

### Official Review · Reviewer_kfdu · 2025-10-31

**Soundness:** 2
**Presentation:** 3
**Contribution:** 1
**Rating:** 2
**Confidence:** 3

**Summary:**

This work introduces **DAG-MoE**, a neural architecture derivative of **Sparse Mixture of Experts (SMoE)**.
The design is based on the observation that aggregating expert outputs in a more complicated way than simple summation can be more expressive.

The authors design a **DAG learning module** where input tokens get routed to their chosen experts like in standard SMoE, after which:

- Expert outputs become initial node embeddings.
- For multiple iterations (depths), the module:
  - Normalizes and projects node features into lower-dimensional space.
  - Learns edge strengths between nodes using learned weights and nonlinear activation.
  - Aggregates information through learned sparse edges (like message passing in GNNs).
  - Adds residual connections for stability.
- After *L* iterations, final node outputs are summed to form the MoE layer output.

The design is compared against SMoE baseline on language pretraining (perplexity) and downstream tasks after fine-tuning.

The paper shows DAG-MoE can theoretically simulate multi-step reasoning (like dynamic programming) within a single layer—something standard SMoE cannot do.

**Strengths:**

- Well-positioned in literature as a natural next step after tree aggregation
- Presence of theoretical analysis of DAG-MoE expressivity which shows that DAG-style aggregation is more expressive for DP algorithmic tasks
- Presence of empirical comparison to SMoE baseline on both perplexity and downstream tasks

**Weaknesses:**

The main weakness is that it's not clear that it's worth doing DAG aggregation instead of the simpler alternative, if we match for wall time. The perplexity differences are fairly small, and we do not see how this scales with training horizon (scaling with model size is definitely a plus!). Downstream performance is very close to random. The structure of DAG computations does not immediately lend itself to very efficient GPU implementations. More precisely:

**Modest improvements at scales tested:** Perplexity improvements are small (up to 0.5 on scale of 13-15). Adding equal extra parameters as shared expert instead of the proposed method is competitive, especially considering no communication requirements in distributed setting. Downstream task performance is very close to random for almost all tasks for both baseline and proposed method.

**Lack of empirical support for reasoning claims:** The paper makes theoretical claims about multi-step reasoning and dynamic programming (Theorem 3.3), but provides no dedicated empirical validation. The modest perplexity gains don't isolate expressivity as the improvement source.

**Missing architectural ablations:** The paper varies capacity (*L* and *d_g*) but doesn't compare against simpler alternatives: fully connected DAG, tree aggregation or shallow MLP aggregator with similar compute cost. The necessity of the DAG structure remains theoretically motivated yet a good understanding of whether DAG structure helps is missing.

**Missing systems analysis:** The DAG module runs sequentially over *L* iterations, which means that in a distributed setting, a large number of communication calls are required, hurting throughput. Moreover, the paper provides FLOPs estimates but no wall-clock timing, throughput, or memory measurements. Without empirical efficiency data, it's unclear whether DAG-MoE's gains justify its added complexity in real distributed settings.

**Questions:**

**1. Proposed experiments for multi-step reasoning claims:**

The paper claims DAG-MoE can simulate dynamic programming within a single layer (Theorem 3.3) but provides no empirical validation. Suggested experiments:

- **Longest Increasing Subsequence (LIS)**: The paper uses LIS as their running example (Appendix A.2.2, Figure 5). Train both MoE and DAG-MoE on synthetic LIS problems of varying lengths. If the theory holds, DAG-MoE should solve longer sequences in fewer layers.

- **Algorithmic reasoning benchmarks**: Test on established DP tasks from datasets like CLRS-30 (algorithmic reasoning benchmark with DP problems) or simple DP tasks: edit distance, knapsack, longest common subsequence.

- **Layer depth ablation**: Compare how many transformer layers MoE vs DAG-MoE needs to solve the same DP problem. The theory predicts DAG-MoE needs fewer layers due to multi-step reasoning per layer.

- **Controlled comparison**: Fix problem complexity, vary model depth. If DAG structure enables multi-step reasoning, shallow DAG-MoE should match deeper standard MoE.

Without experiments of this kind, the DP/reasoning claims remain purely theoretical.

**2. How do you parallelize the sequential DAG aggregation?**

**3. The paper would benefit from ablations comparing DAG structure against simpler aggregation alternatives (tree, dense, MLP)?**

**4. What are the wall-time and peak memory costs compared to baseline MoE?**

**5. The pretraining results in 5.2. control for added parameters - do they also control for FLOPs?**

---

### Official Review · Reviewer_PQXU · 2025-11-01

**Soundness:** 2
**Presentation:** 3
**Contribution:** 2
**Rating:** 4
**Confidence:** 4

**Summary:**

This paper presents DAG-MoE, a interesting Mixture-of-Experts (MoE) architecture designed to improve scalability and expressiveness in MoE-based LLMs. Instead of relying on the conventional weighted-sum aggregation of expert outputs, the authors introduce a structural aggregation mechanism that learns a directed acyclic graph (DAG) among selected experts. This approach expands the expert-combination space and allows potential multi-step reasoning within a single MoE layer, without changing the router or expert configurations. Theoretical analysis supports the expressiveness advantage, and experiments on standard language modeling benchmarks demonstrate consistent performance gains in both pretraining and fine-tuning compared to traditional MoE baselines.

**Strengths:**

1. Well-motivated and innovative idea: The paper is well-motivated, aiming to improve both the scalability and expressiveness of large language models. The proposed structural aggregation mechanism, which learns a directed acyclic graph (DAG) among selected experts, is conceptually interesting and offers a fresh perspective on MoE design.

2. Lightweight and compatible architecture: The proposed DAG-MoE framework is lightweight and can be seamlessly integrated into existing MoE architectures without significant modification or computational overhead, making it practical for real-world deployment.

3. Comprehensive and convincing experiments: The experimental evaluation is extensive and well-structured, validating the proposed method from multiple aspects including effectiveness, generalization, and robustness. Results clearly demonstrate the research value and practical benefit of DAG-MoE.

**Weaknesses:**

1. Unclear motivation for the architectural design: The motivation behind the architecture of DAG-MoE is not clearly articulated. Although several learnable parameter matrices are introduced, the paper does not sufficiently explain the rationale for this specific design choice or how it contributes to the intended structural aggregation.

2. Limited theoretical depth: While the paper provides extensive analysis and includes proofs in the appendix, the theoretical discussion lacks depth. It does not offer a fundamental or intuitive explanation that convincingly supports the soundness and necessity of the proposed method.

3. Imbalanced structure and presentation: The overall organization of the paper is somewhat unbalanced. The method section is overly detailed, whereas the experimental section is relatively brief, which weakens the empirical narrative and makes it harder to assess the practical impact of the approach.

4. Insufficient experimental scale and ablation studies: Although the paper covers multiple evaluation aspects, each experimental setting includes only a limited number of cases. The lack of large-scale experiments and comprehensive ablation studies makes it difficult to fully validate the effectiveness and robustness of DAG-MoE.

5. Missing efficiency analysis: DAG-MoE is likely to introduce additional computational overhead due to its structural aggregation mechanism, but the paper provides no analysis or discussion on efficiency or scalability trade-offs.

**Questions:**

1. Could the authors provide a more fundamental theoretical justification or analytical result to support the soundness and necessity of the proposed method?

2. Could the authors elaborate on the motivation behind the architectural design of DAG-MoE, particularly why the introduced learnable parameter matrices are structured in this way?

3. Could the authors include additional experimental results — such as evaluations on more recent backbone models, comparisons across different model scales, and more comprehensive ablation studies — to further substantiate the method’s effectiveness?

4. Could the authors provide a detailed efficiency analysis, including computational cost and scalability, and compare it with prior works that modify the router or increase the number of experts?

---

### Meta-Review · Area_Chair_qbkr · 2026-01-12

**Summary:**

The paper proposed a DAG-based, message-passing style aggregation of experts to replace the weighted summation in current Mixture-of-Experts (MoE) models. Such DAG-MoE only requires lightweight additional training of a few projection matrices, but its structural aggregation can conduct multi-step reasoning and dynamic programming that cannot be directly handled by expert summation. They provide both theoretical analysis and empirical experiments to justify the advantage of DAG-MoE in expressiveness, perplexity, downstream performance, and efficiency.

**Reviewer Concerns:**

- Although most reviewers found the idea of structural aggregation in DAG-MoE to be novel, they are not impressed by the advantages brought by the method: the improvement on both perplexity and downstream task performance is modest, and the sequential inference of DAG-MoE may affect the latency, throughput, and wall-clock time cost.
- Theoretical analysis lacks depth and is incomplete. The multi-step reasoning and dynamic programming claims are only surface-level, lacking detailed empirical validation on more algorithmic datasets.
- Lack of analysis or interpretation of the algorithmic structures implemented by the DAG-MoE learned for different tasks.
- Lack of comparisons with simpler baselines, such as MLP aggregation and fully-connected DAG: this makes the necessity of developing complex DAG-MoE questionable.
- Lack of large-scale experiments.
- Missing comparison with contemporary MoE systems and other existing structural aggregation methods of experts, like the chain-of-experts and S'MoRE.
- Lack of discussion on the strong structural constraints and hyperparameter sensitivity in the empirical implementation.

**Reviewer Scores:**

The initial review scores are 6, 4, 4, 2, 2. Since the authors have not submitted any responses, there is no further feedback from the reviewers. I checked the review comments and found several major concerns shared across reviewers. The weaknesses pointed out by the only positive reviewer also focus on the poor interpretability of the algorithmic structure/patterns discovered by DAG-MoE, as well as the lack of a necessary ablation study. I think the idea of the paper has potential, but the current draft does not provide convincing analysis and empirical results.

---

### Decision · Program_Chairs · 2026-01-26

Reject